# Canonical goal-selective representations are absent from prefrontal cortex in a spatial working memory task requiring behavioral flexibility

**Claudia Böhm\*, Albert K Lee\***

Howard Hughes Medical Institute, Janelia Research Campus, Ashburn, United States

**Abstract** The prefrontal cortex (PFC)'s functions are thought to include working memory, as its activity can reflect information that must be temporarily maintained to realize the current goal. We designed a flexible spatial working memory task that required rats to navigate – after distractions and a delay – to multiple possible goal locations from different starting points and via multiple routes. This made the current goal location the key variable to remember, instead of a particular direction or route to the goal. However, across a broad population of PFC neurons, we found no evidence of current-goal-specific memory in any previously reported form – that is differences in the rate, sequence, phase, or covariance of firing. This suggests that such patterns do not hold working memory in the PFC when information must be employed flexibly. Instead, the PFC grouped locations representing behaviorally equivalent task features together, consistent with a role in encoding long-term knowledge of task structure.

**\*For correspondence:**
boehmc@janelia.hhmi.org (CB);
leea@janelia.hhmi.org (AKL)

**Competing interests:** The authors declare that no competing interests exist.

## Introduction

Animals can pursue a goal while handling distractions and delays, starting from a variety of initial conditions, and adapting their responses in the face of unexpected obstacles. To guide such flexible goal-directed behavior, there needs to be a representation of the goal itself that is robust to these circumstances, which can provide top-down instruction for selecting appropriate actions. The prefrontal cortex (PFC) plays a central role in flexible goal-directed behavior (*Miller and Cohen, 2001*; *Fuster, 2015*), and one of its primary functions is thought to be the maintenance of information relevant for achieving the current goal (i.e. working memory) (*Fuster and Alexander, 1971*; *Funahashi et al., 1989*; *Miller et al., 1996*; *Rainer et al., 1998*; *Romo et al., 1999*; *Wang, 1999*; *Erlich et al., 2011*; *Wimmer et al., 2014*; *Inagaki et al., 2019*; *Wu et al., 2020*). Therefore, the PFC is a prime candidate area for containing a representation of the current goal itself.

Spatial working memory tasks are well-suited for investigating such representations. First, the current goal, a particular spatial location, can be clearly specified and is ethologically relevant for many species (*O'Keefe and Nadel, 1978*). Second, all of the aspects of flexibility mentioned above can be incorporated naturally in a form that many animals, including rodents, can solve. However, rodent spatial working memory experiments employed to date with recording in PFC or other brain areas (*Wood et al., 2000*; *Frank et al., 2000*; *Baeg et al., 2003*; *Fujisawa et al., 2008*; *Gill et al., 2011*; *Harvey et al., 2012*; *Pfeiffer and Foster, 2013*; *Wikenheiser and Redish, 2015*; *Ito et al., 2015*; *Spellman et al., 2015*; *Kim et al., 2016*; *Guise and Shapiro, 2017*; *Bolkan et al., 2017*) have not combined all of these elements of flexibility in a single task. As a result, potential working memory representations of the current goal have been difficult to dissociate from behavioral or sensory correlates. For instance, classic T-mazes have a single start and single route to each of the two goals.

Thus, differential neural activity before reaching the T junction could represent the goals themselves, the two sets of stereotyped actions used to reach each goal as well as any associated sensory correlates (e.g. looking left versus right), or the plan to 'go left' or 'go right'. Furthermore, any goal-specific activity could differ if the animal were to start from another point, and therefore not be usable under different initial conditions. Here, we devised a novel spatial working memory task incorporating multiple aspects of behavioral flexibility, allowing a search for a 'pure' representation of the goal itself by disambiguating goal-related activity from other correlates.

## Results

In our task, rats needed to remember one of three goal locations in each trial. The current goal was encoded during a 'sample phase', in which animals were guided with light cues to one goal where they received a small reward. Rats had to remember this location until they needed to navigate to that goal again in the 'test phase' – starting from one of three different locations and via one of three routes in the absence of lighted cues – to receive a large reward. The different routes were implemented by using an elevated maze design with bridges that could be raised (open) and lowered (blocked). The available route was only revealed after a 3 s (3.2 s in one animal) 'fixation' period (hereafter referred to as the 'delay period') during which animals had to hold their nose in one of the three test phase 'start' location ports. The correct start location port was assigned pseudorandomly in each trial. To find the correct start port between each trial's sample and test phase, animals had to poke their nose into different ports until they found the one that elicited a tone when poked, which indicated the correct choice (*Figure 1A,B*, *Video 1*). The goal sample phase route and test phase route were also pseudorandomly assigned in each trial (see Materials and methods). This design requires animals to update the currently relevant goal every trial (working memory task) and pushes them to remember the goal itself instead of memorizing a specific behavioral sequence or planning a particular motor action to reach the goal. Thus, the goal location is the key variable to retain, which must then be used flexibly to solve the task: navigating to that goal from any location by any route. Furthermore, having more than two goals excludes a strategy of navigating to one goal by default and only remembering when the goal is the other one – that is having three goals promotes the use of working memory representations of each goal itself.

Animals reached high levels of performance (*Figure 1C,D*, mean performance over all rats: 77.37, 95% confidence interval [CI]: [72.46, 81.97]). To examine whether animals indeed remembered the goal location itself instead of particular routes, we compared the performance in trials where the route (bridge) animals had to take in the test phase was the same or different from the outbound and/or return bridge in the sample phase. The performance was comparable across routes for all animals, indicating that rats remembered goal locations instead of routes (*Figure 1D*, left). In trials where the goal was adjacent to (i.e. 60 degrees in either direction from) the available route, rats most often took the shortest route (88%, 81%, and 95% of trials for each animal, respectively), supporting the idea that animals used a spatial map instead of a fixed route plus recognition strategy. Furthermore, the performance was comparable and above chance for all goal locations, implying that rats remembered each goal location rather than relying on a strategy of remembering only a subset of them (*Figure 1D*, center left). Rats spent several seconds in each task phase and generally ran faster on test outbound runs than sample outbound runs (*Figure 1D*, center right). The search time to find the correct start location was on average slightly higher in the trials where the subsequent choice was incorrect, but overlapped with search times in correct trials (*Figure 1D*, right). During the test phase, rats rarely stopped at the end of the available bridge after crossing it on their way to the goal in a manner that might reflect vicarious trial and error behavior at the choice point (*Johnson and Redish, 2007*) (see Materials and methods for details).

A Neuropixels probe (*Jun et al., 2017a*) was chronically implanted in medial prefrontal cortex (mPFC), previously shown to be required in variants of simpler rodent spatial working memory tasks (*Spellman et al., 2015*; *Guise and Shapiro, 2017*). One hundred to two hundred units were simultaneously recorded across subareas including anterior cingulate, prelimbic (PL), infralimbic, and dorsal peduncular cortices (*Figure 1B*) (n = 3 animals: 182, 131, and 98 cells in the three main sessions analyzed, one from each animal; in addition, for two animals, one novel rotation experiment session each, consisting of 152 and 186 cells, was used for the analysis of relearning). We primarily focused our analyses on the delay period since, during that time, behavior is well-controlled, and animals

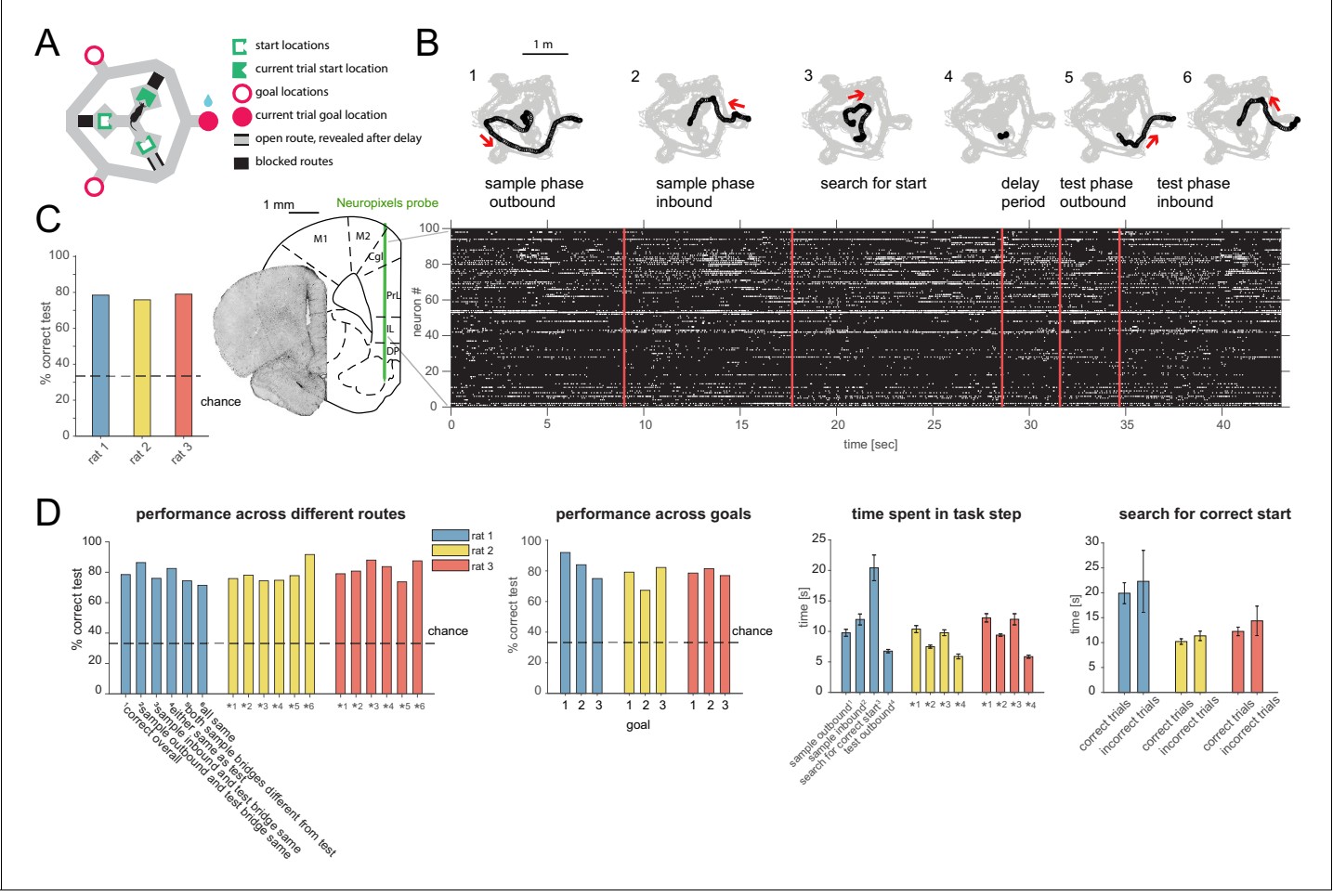

**Figure 1.** Task design, behavior and recording. (**A**) Schematic of 'multi-start/multi-goal/multi-route' (MSMGMR) task environment. (**B**) Top: Each trial consisted of the following steps: (1) the current goal is randomly assigned and cued with lights; a single, randomly assigned route (bridge) is available; rat gets small reward upon arrival at cued goal; (2) animal returns to center via any route; all routes are blocked upon arrival at center; (3) animal searches for randomly assigned start position, indicated by a tone once animal pokes nose into correct port; (4) animal must maintain nose poke for 3 s (3.2 s in one animal); (5) a randomly assigned route becomes available and animal can navigate to goal; and (6) animal returns to center via any route to initiate next trial (see *Video 1*). Bottom left: Neuropixels probe is chronically implanted in mPFC, and 384 channels spanning multiple subareas are recorded from simultaneously. Bottom right: Spiking activity during task. (**C**) Task performance (test phase). (**D**) Left: Task performance in subsets of trials in which different routes were taken. Again, a single outbound route was randomly assigned for both test and sample phases; inbound routes could be freely chosen by the animal. Center left: Performance for each goal location. Center right: Time spent for each of the task steps (see also B) (mean and 95% CI). Right: Search time for the start location for correct and incorrect trials (mean and 95% CI).

must remember a given goal while being in different defined locations and facing different defined directions, as well as not knowing the required future motor action. Analyses were applied to all putative principal cells with stable firing rates throughout the session (97, 84, and 68 cells in the three main sessions, and 84 and 105 cells in the additional relearning sessions; see Materials and methods) and pooled across subareas (results including all cells or split by subarea were similar and are provided in the supplementary figures as indicated).

We searched for representations of the current goal encoded in terms of the major forms of delay period activity previously found in other working memory tasks, spatial or non-spatial, involving recordings from the PFC or elsewhere in primates and rodents: activity reflecting the representation at the time of encoding the sample item (*Funahashi et al., 1989*; *Miller et al., 1996*; *Rainer et al., 1998*; *Romo et al., 1999*; *Wu et al., 2020*), elevated/suppressed activity in single cells (*Fuster and Alexander, 1971*; *Funahashi et al., 1989*; *Miller et al., 1996*; *Rainer et al., 1998*; *Romo et al., 1999*; *Kim et al., 2016*; *Inagaki et al., 2019*), or sequential firing patterns across multiple cells that tile the delay period (*Baeg et al., 2003*; *Fujisawa et al., 2008*; *Pastalkova et al., 2008*;

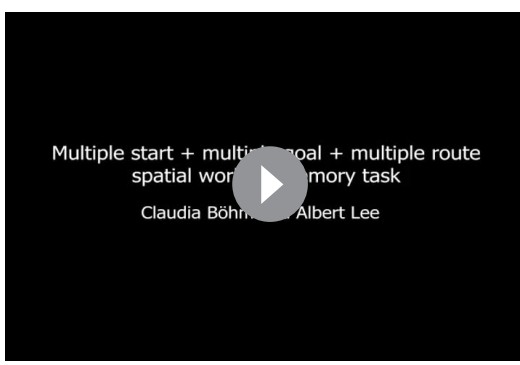

**Video 1.** Video showing three consecutive trials of animal performing the task.
https://elifesciences.org/articles/63035#video1

*Harvey et al., 2012*; *Ito et al., 2015*), oscillatory phase-dependent firing (*Siegel et al., 2009*; *Watrous et al., 2018*), and elevated/suppressed covariances in firing among pairs of neurons (*Barbosa et al., 2020*).

We first tested whether memory of the current goal could be maintained during the delay period by a firing rate pattern across cells (population vector, PV) similar to the one when the animal was at the goal itself during the sample phase. To begin with, the PV at each goal during the sample phase was distinguishable and stable over time (*Figure 2A*, *Figure 2—figure supplement 1A*), including across sample and test phases (*Figure 2—figure supplement 1B*). In addition to distinguishing which goal the animal was at in both sample and test phases, the task phase could also simultaneously be almost perfectly decoded (*Figure 2—figure supplement 1B*, right), likely because the animal received less reward in the sample phase. Having established the presence of a stable spatial representation at each goal, we correlated the overall activity during the delay period with the sample phase PV at each goal and asked whether it was more correlated with the currently remembered goal's PV. This was not the case (*Figure 2B*), as also seen in a different task (*Guise and Shapiro, 2017*). To test whether the remembered goal was represented as transient increases in correlation or a switching between current and other goals with the relevant (current) one being overrepresented (*Kelemen and Fenton, 2010*), we attempted to predict the goal based on correlation scores resulting from sliding a window of variable width across the delay period. While the animal's current (i.e. start) location was readily decodable (as expected), the remembered goal was not, using a wide range of time bins (*Figure 2C*) or activity restricted to individual subareas (*Figure 2—figure supplement 2*) or to cells with significant spatial selectivity at goals (*Figure 3—figure supplement 1*; note that individual cells in any subarea could show spatial selectivity at goals, as well as at starts, at both, or neither).

If the remembered goal is not maintained by activity directly related to activity at the goal itself, it could be (1) transformed into a different, but goal-specific, pattern, potentially dependent on the start location, (2) encoded in egocentric coordinates (i.e. the direction relative to the current start location; *Sarel et al., 2017*) instead of in terms of the absolute (allocentric) goal location, (3) represented by a sequential, instead of tonic, activity pattern, and/or (4) reflected in the phase of spike times or the short timescale interactions between pairs of neurons (*Barbosa et al., 2020*). We initially tested if any single-cell activity in 100-ms- to full-delay-period-sized time bins showed consistent firing differences for allocentric, start-dependent, or egocentric goal location (*Figure 3—figure supplement 2*). We found significant differences for the current (i.e. start) location, but not allocentric or egocentric goal location. There was also no evidence of start-dependent encoding of goals, that is a unique code for the nine start-goal pairs.

We then tested whether the remembered goal was encoded with a sequential activity pattern across multiple cells that may not be detectable at the single-cell level (*Figure 3*). We employed several classification methods at multiple time resolutions (*Figure 3C*). Note that, for this analysis, potential activity patterns were always referenced to the delay period onset, as previously seen for memory-related sequences that tile the delay period (*Fujisawa et al., 2008*; *Pastalkova et al., 2008*; *Harvey et al., 2012*). Results were consistent across methods and time resolutions: current location could readily be decoded with any classification method for time bin resolutions between 100 ms to the full delay period duration. In contrast, no method could successfully classify the remembered goal in allocentric or egocentric coordinates at any time resolution (*Figure 3C*), including when using all cells regardless of firing rate stability (*Figure 3—figure supplement 3*).

We considered whether in our well-trained animals the representation of the remembered goal might be less prominent than during learning, and therefore, if goal representations may be more easily identified when the animal is learning (*Liu et al., 2014*; *Maggi et al., 2018*). While we did not

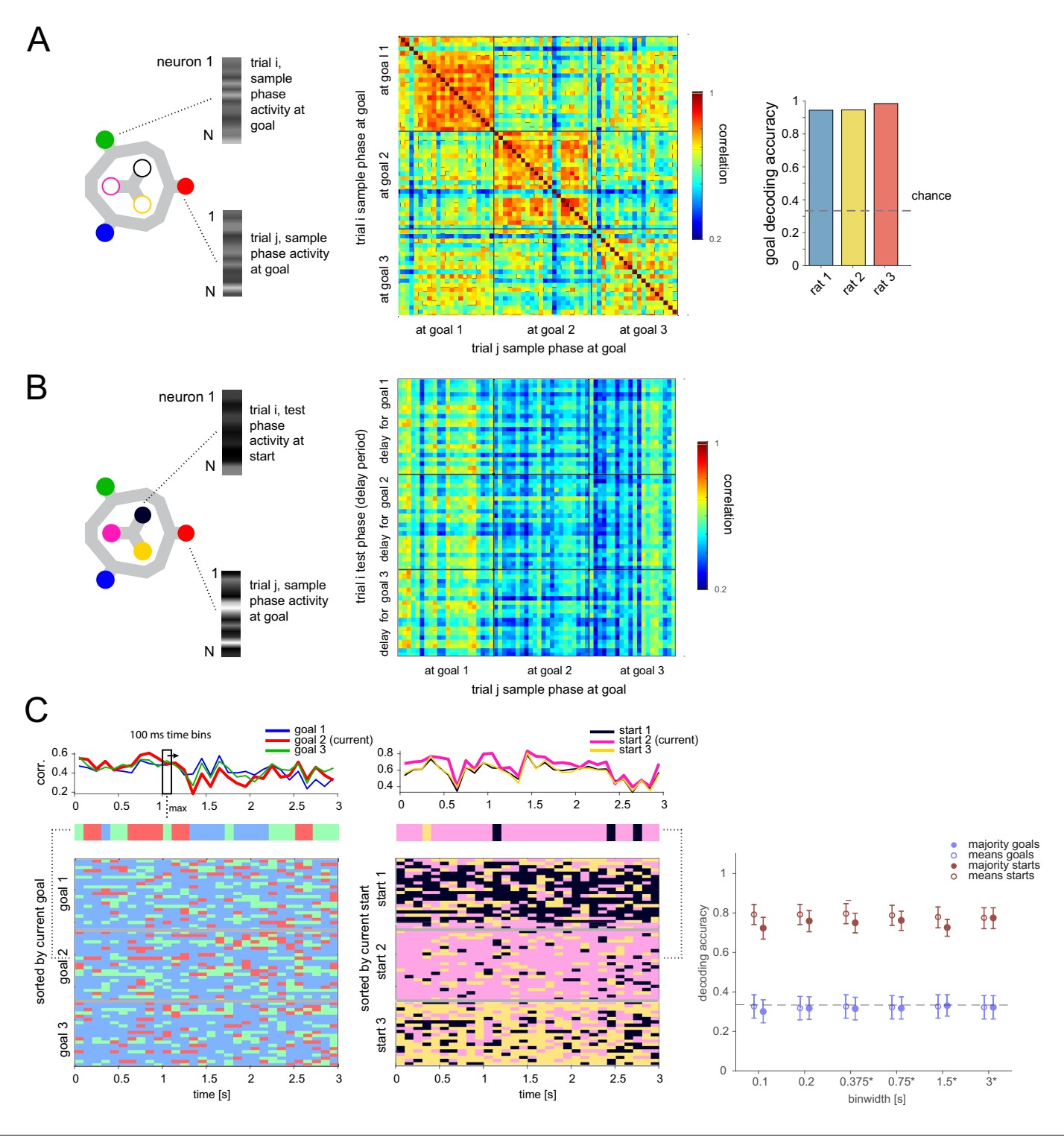

**Figure 2.** Memory is not maintained by goal-location-specific activity in the delay period. (A) Population vectors (PVs) of activity while animal at each goal during sample phase (left) are distinct and stable (middle, correlation matrix of single trials in one animal; right, decoding accuracy, logistic regression classifier, mean over all trials and animals: 94.48%, 95% bootstrap CI: [90.87, 96.80]). (B) Correlation of PV while at goal during sample phase and PV during delay period while animal must maintain that goal-specific information. (C) Left top: Example single-trial PV over time during delay period correlated with each goal PV. Left middle: Goal with maximum correlation at each time bin above. Left bottom: Same for all correct trials in this animal, sorted by current goal. Middle: Analogous to left but correlated with each start location PV (excluding contribution from current trial). Right: Correlation-based classification for range of binwidths (mean and 95% CI). Class per trial determined by highest mean correlation over entire delay

*Figure 2 continued on next page*

*Figure 2 continued*

(unfilled) or majority vote of class with highest correlation at each time point (filled). *Binwidths were 0.4, 0.8, 1.6, and 3.2 s for one animal that had 3.2 (versus 3) s delay (also for *Figures 3C*, *6B*).

The online version of this article includes the following figure supplement(s) for figure 2:

**Figure supplement 1.** Representations are distinct and stable at goals.

**Figure supplement 2.** Goal-location-specific representations are not maintained in the delay period for any subarea.

record activity during the long training period, we performed a behavioral manipulation experiment after the task had been learned in which animals (n = 2 sessions from two animals) had to relearn the task in an altered configuration. Specifically, we rotated the maze by 60 degrees ~ 1/3 of the way through the recording session, so that the reference frame was changed, with the goal and start positions now in between their previous positions (*Figure 4A*). Animals had never seen this configuration before or seen a rotation of the maze. Consistent with the idea that the animals must adapt to the new configuration and learn to apply previously internalized rules, performance dropped dramatically then gradually improved over the remainder of the session (*Figure 4A*). However, during this relearning, when the cognitive demand might be higher and thus elicit a more prominent representation of critical task variables, population analysis still could not decode the currently remembered goal during the delay period. In contrast, spatial representation of the start locations was again clearly represented even as the animal adapted to the new configuration (*Figure 4B,C*).

We next tested whether information might be stored in spike timing relative to local field potential (LFP) oscillations (*Siegel et al., 2009*; *Watrous et al., 2018*). We explored three frequency bands (2.5–5 Hz, 5–12 Hz, and 15–30 Hz), identified based on their elevated power in the delay period (*Figure 5A*, left). First, we calculated each cell's goal-specific phase preference in the delay period. We compared the distribution of phase preference magnitudes to one where goal labels were shuffled. The distributions were not different, neither when including all stable cells, nor when selecting only cells that showed significant overall phase locking, suggesting the remembered goal does not affect overall phase preference (*Figure 5A*, *Figure 5—figure supplement 1A,B*). Second, we asked whether spike counts at specific phases might differ in a goal-specific manner, either when using all stable cells or only those that were significantly phase locked to at least one of the goals, but they did not (*Figure 5A*, *Figure 5—figure supplement 1C*). We also explored the possibility of a recently described form of 'activity-silent' memory, in which working memory is expressed in the spiking synchrony between pairs of neurons while stimulus information is not decodable from firing rates (*Barbosa et al., 2020*). However, neither for the pairs of neurons exhibiting excitatory interactions nor for the pairs of neurons exhibiting inhibitory interactions did covariances differ between trials associated with one goal versus the others (*Figure 5B*, *Figure 5—figure supplement 2*, see Materials and methods). Together, these results suggest that previously described forms of working memory maintenance are not responsible for storing the current goal in our task in which this information must be employed flexibly.

To test whether goal representations are present in other periods than the (nosepoke 'fixation') delay period, we analyzed data in a time-resolved fashion aligned to multiple time points during trial progression. Specifically, we tested how well the current goal could be decoded with respect to key behavioral reference time points: (1) when the animal arrives at the goal during the sample phase, (2) when the animal returns to the center during the transition between sample and test phases, (3) when the route becomes available, (4) at the choice point when the animal enters the outer ring of the maze in the test phase, and (5) when the animal arrives at the goal during the test phase. Importantly, we performed these analyses not only using the neural data but also, separately, using the position tracking data of the two LEDs on the animal's head. We found that it was possible to decode the currently remembered goal location from the neural data at various time points. However, the remembered goal could also be decoded from the behavioral tracking data alone with a very similar time course. Thus, it is likely that this neural representation of the currently remembered goal at these times is due to the animal's position, orientation, posture, or other behavioral features (*Figure 3—figure supplement 4*). Crucially, the decoding performance based on tracking data was at chance levels throughout the nose poke delay period (*Figure 3—figure supplement 4*, third row,

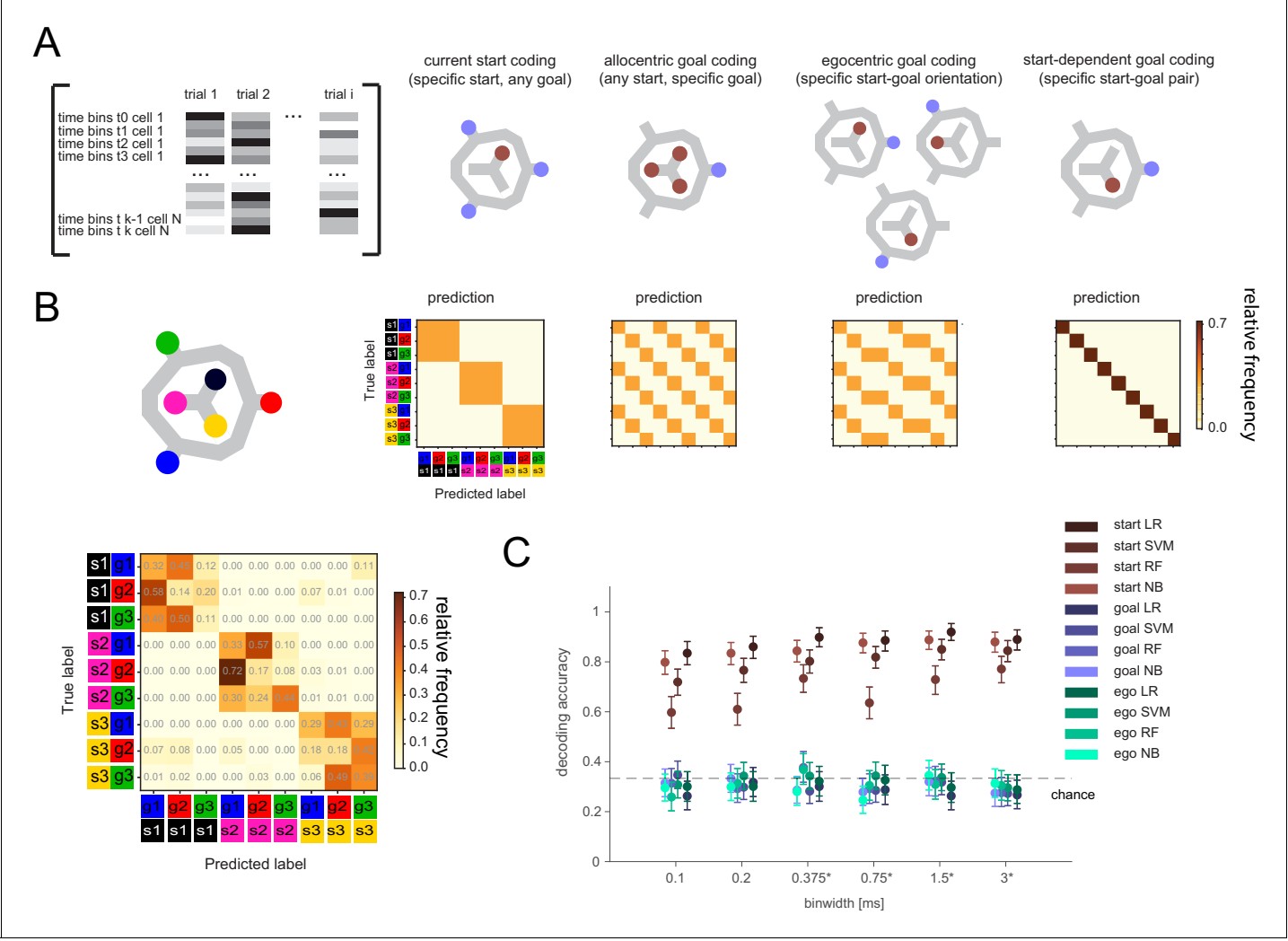

**Figure 3.** Lack of differential activity patterns corresponding to the current, remembered goal in the delay period. (**A**) Leftmost: Differential patterns considered corresponded to activity across cells and time bins during delay period. Potential encoding schemes (left to right): start location represented independently of current, remembered goal; current goal represented independently of current (start) location; current goal represented in egocentric coordinates, that is direction to current goal with respect to current (start) location; current goal represented distinctly in different start locations. (**B**) Population activity analysis of potential encoding schemes during delay period using supervised classification. Top: Confusion matrices expected for each scheme. Bottom, left: Confusion matrix using support vector machine (SVM) classification (0.75 s bins) for one animal. (**C**) Three-class delay period activity classification using logistic regression (LR), SVM, random forest (RF), or Naïve Bayes after feature selection (NB) over range of time resolutions (mean and 95% CI).

The online version of this article includes the following figure supplement(s) for figure 3:

**Figure supplement 1.** Local spatial selectivity of individual cells at goal and/or start locations.

**Figure supplement 2.** Single-cell firing rate analysis for individual animals.

**Figure supplement 3.** Population decoding for all cells independent of selection criteria.

**Figure supplement 4.** Decoding of current goal during task progression.

right, period from −3 to 0 s), providing direct evidence that our task design reduces the presence of confounding behavioral variables during that period.

If the mPFC does not encode memory of the goal in this task, what task-relevant processes might it support? We tested whether other task features could be decoded from mPFC activity. First, we analyzed if the activity at the goal in the sample phase (presumably during encoding) differs when there is an error in the subsequent test phase or not, but this was not the case. In contrast, after the animal had made an incorrect choice in the test phase, activity at the goal was markedly different,

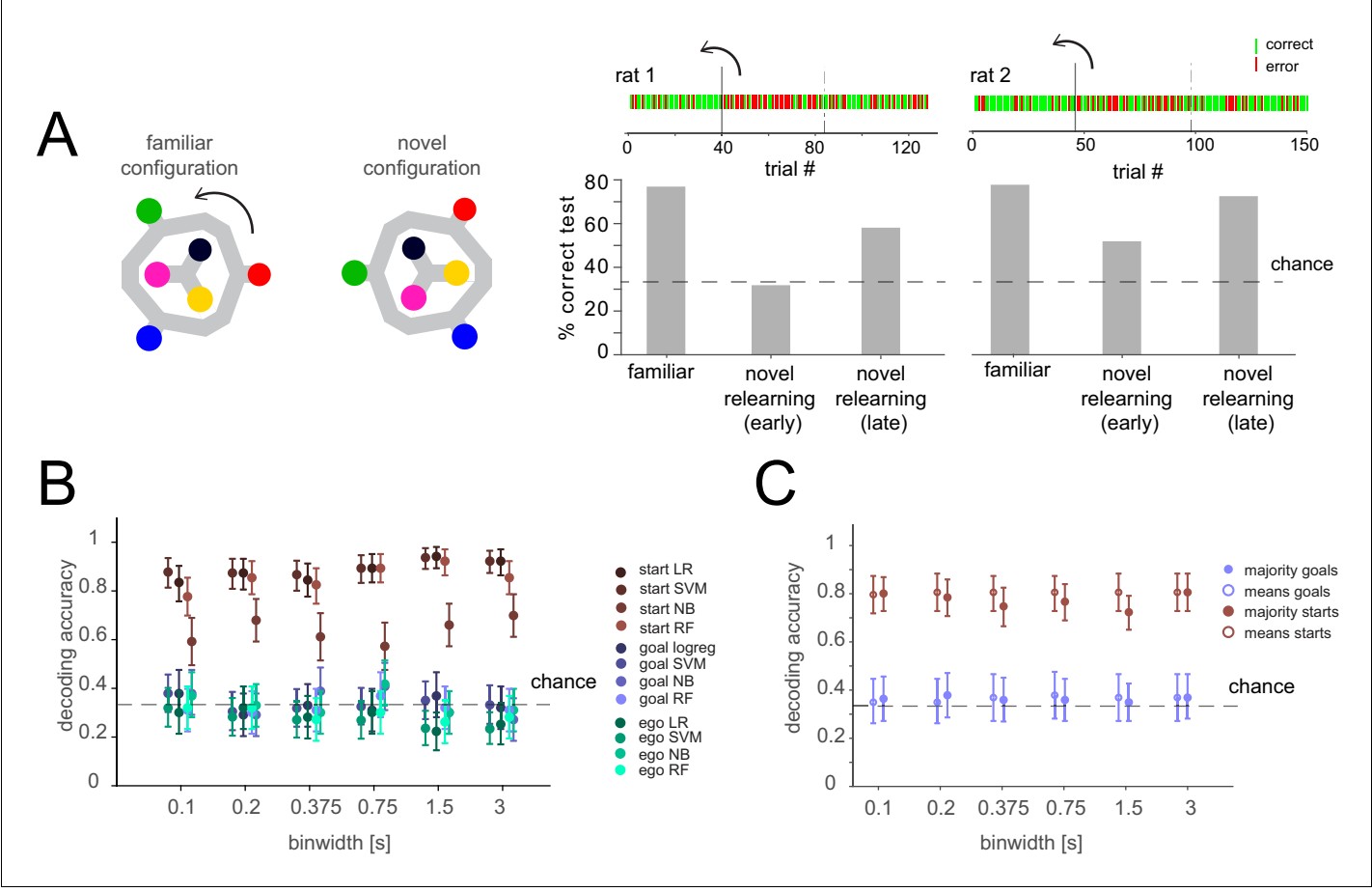

**Figure 4.** Lack of differential activity patterns corresponding to the current, remembered goal in the delay period during relearning. (**A**) Left: Task layout in familiar and novel configuration. The maze was turned by 60 degrees after 40 or 46 trials for the two animals, respectively. Right: Performance before the rotation (familiar), after the rotation (novel relearning – early) and in the last ~1/3 of the session (novel relearning – late). Above, outcomes of single trials are shown (green: correct, red: error, solid line indicates time of rotation, dashed line indicates division of trials in early and late relearning periods). (**B**) Three-class delay period activity classification using logistic regression (LR), SVM, random forest (RF), or Naïve Bayes after feature selection (NB) over range of time resolutions (mean and 95% CI). Same analysis as *Figure 3C*, but for relearning trials (all trials after the rotation). (**C**) Correlation-based classification for range of binwidths (mean and 95% CI). Same analysis as *Figure 2C*, right for relearning trials.

presumably due to the lack of reward; however, correlations between the PVs of activity at the goal among correct trials and, separately, among error trials was comparable (*Figure 6A*). Furthermore, mPFC delay period activity did not indicate an upcoming or past error (*Figure 6B*), further corroborating that mPFC might not directly store current memory content. We then checked if mPFC distinguished the two task phases not only at the goal (*Figure 2—figure supplement 1B*), which could be due to the amount of reward but when animals returned to the center. Before reaching the center, and at the center, task phase was not decodable (*Figure 6C*) (note the decodability afterwards could arise from cue or behavior differences in the two phases that were not present earlier). Lastly, we compared the population activity while rats engaged in different behaviors at different locations. Within each group of behaviors (i.e. waiting at a start nose port, crossing a bridge/route, consuming reward at a goal), mPFC displayed spatial selectivity (e.g. it differentiated the three bridges). Furthermore, we found that this spatial selectivity was embedded within a larger organization of activity in which these distinct, task-relevant groups were clearly separable from each other (*Figure 6C*, *Figure 6—figure supplement 1*).

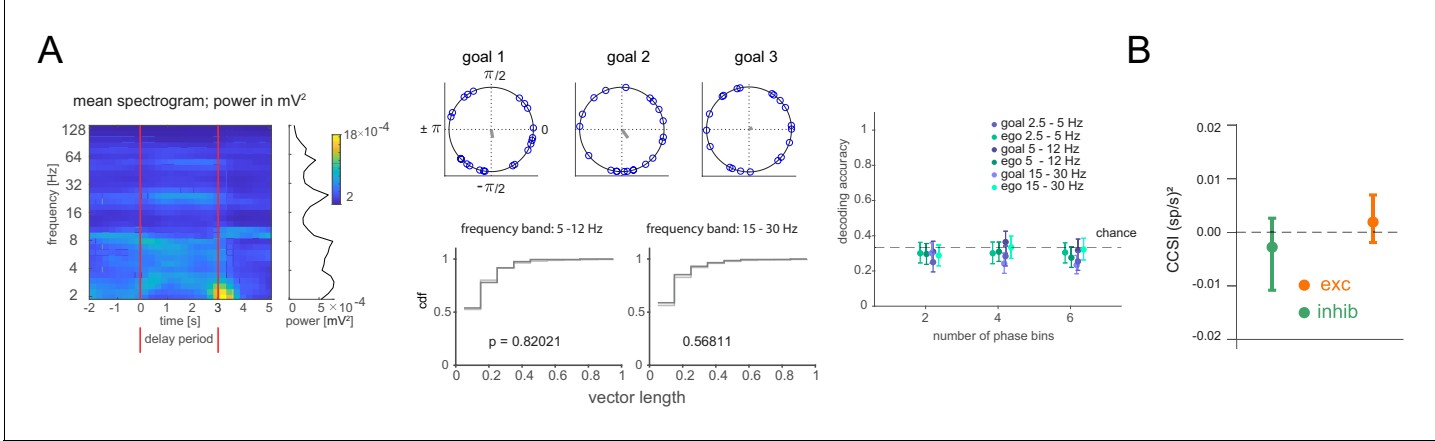

**Figure 5.** Lack of differential phase or covariance of firing corresponding to the current, remembered goal in the delay period. (**A**) Phase analysis. Left: Spectrogram of delay period local field potential (LFP) with average power during the delay period at right (LFP was notch-filtered at 60 Hz, then power was computed within each of 31 logarithmically spaced bins). Middle: Example cell phase preference of delay period spikes and resultant vector length (r, gray) across all trials for each current goal (top). Cumulative distribution of r for all cells from one animal compared to shuffle of trials for two frequency bands (bottom). Right: Decoding accuracy (mean and 95% CI) using spike counts at specific phases. Phases for each frequency band were divided into 2, 4, or 6 phase bins. (**B**) cross-correlation selectivity index for the delay period (CCSI, after *Barbosa et al., 2020*) is a measure of the difference in covariance between trials where the current goal is the one where a given pair of neurons preferentially fires at during the sample period and trials where the current goal is either of the other two goals for cell pairs determined to have excitatory or inhibitory interactions (mean and 95% CI, see Materials and methods).

The online version of this article includes the following figure supplement(s) for figure 5:

**Figure supplement 1.** Phase analysis for individual animals.
**Figure supplement 2.** Covariance analysis for individual animals.

## Discussion

Previous work has shown sensory stimulus-specific delay period activity (delay activity) independent of motor plans (*Romo et al., 1999*; *Wu et al., 2020*) or resistant to distractor stimuli (*Miller et al., 1996*), and start-independent spatial (*Brown et al., 2016*; *Guise and Shapiro, 2017*; *Watrous et al., 2018*) or route-independent visuospatial (*Saito et al., 2005*) goal-specific delay activity in the PFC. A pair of studies (using a single start location; *Spellman et al., 2015*; *Bolkan et al., 2017*) showed no evidence of goal-selective delay activity in rodent PL mPFC. In one of these studies, motor planning was prevented, but not in the other (i.e. standard T-maze), and a study similar to the latter one (*Kim et al., 2016*) found goal-selective delay activity. Another study (*Lara and Wallis, 2014*) found no evidence of visual stimulus-selective delay activity in primate PFC. However, this study used color as the relevant stimulus dimension and found little evidence of color-selective activity even during stimulus presentation. Since PFC neurons have been shown in other cases to encode sample stimulus color (*Buschman et al., 2012*), this suggests that encoding of the stimulus during the sample period may be a prerequisite for observing stimulus-selective activity in the delay period. Spatial information is strongly represented in PFC in primates (*Funahashi et al., 1989*; *Rainer et al., 1998*; *Saito et al., 2005*; *Lara and Wallis, 2014*) and essentially all rodent studies, including here (*Figure 2A*, *Figure 2—figure supplement 1*, *Figure 3—figure supplement 1*); yet, we found no spatial goal-selective delay activity. Furthermore, our ability to decode the current start position and also the goal at various time points during the task (*Figure 3—figure supplement 4*), and the relatively high number of cells recorded simultaneously in our study, suggests that our data set was large enough to have detected an effect of sizes previously reported in the literature for simpler tasks.

In contrast to previous studies, we combined all elements of flexibility in one task – distractions, different start locations, and different unpredictable routes, as well as more than two goals. We found no evidence of goal-selective delay activity in fully trained animals or during relearning in any of the major forms previously documented over a wide range of parameters and across large numbers of simultaneously recorded cells in multiple mPFC subareas. Thus, these representations are

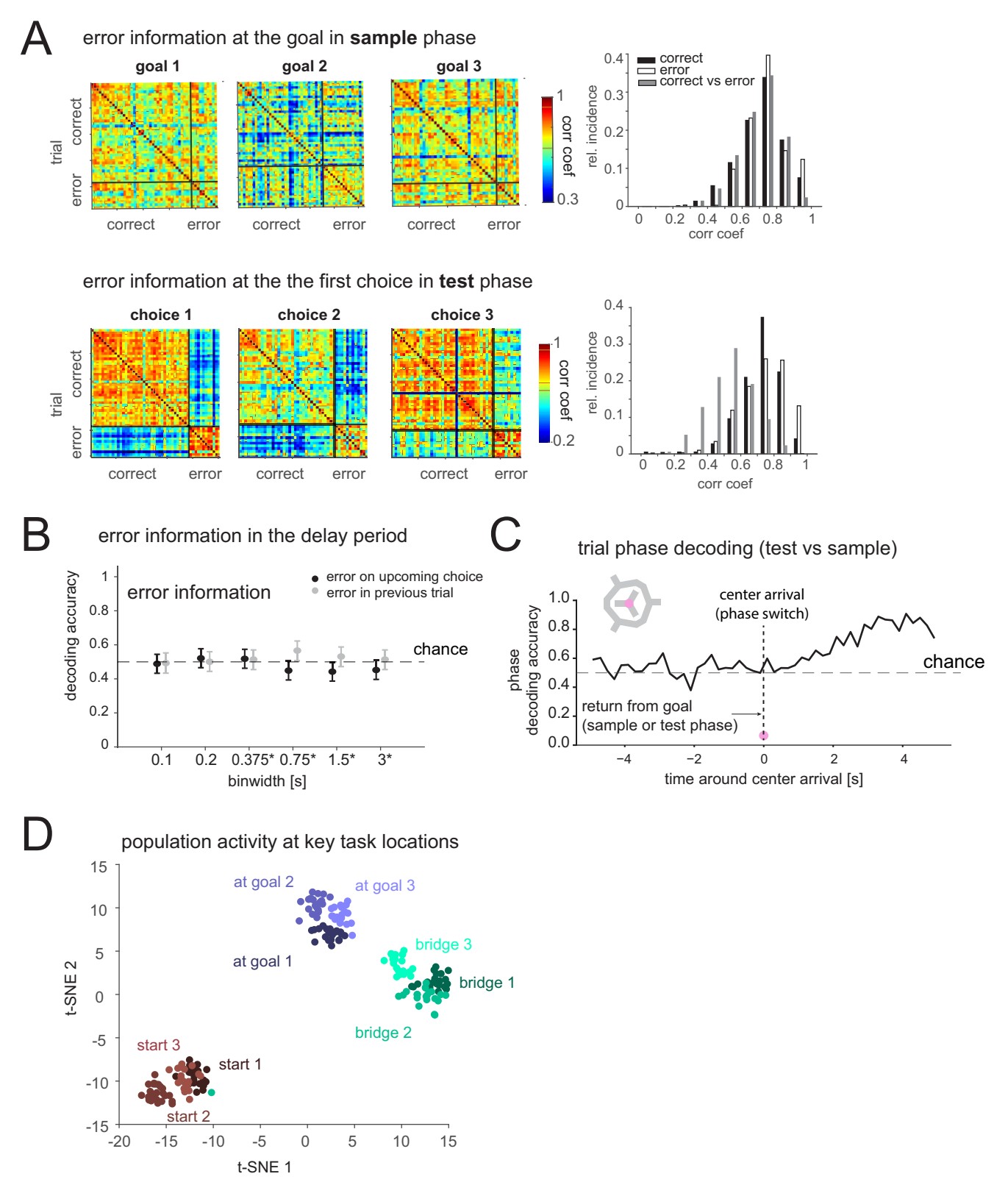

**Figure 6.** Prefrontal cortex encodes task-relevant information and forms groups of behavioral equivalence. (**A**) Top: Correlation matrices between population vectors of activity at each goal in sample phase of trials where the animal is correct or incorrect in the subsequent test phase. Distribution of correlation coefficients from these matrices (right). Bottom: Correlation matrices of population vectors at each first choice goal location in test phase for

*Figure 6 continued on next page*

*Figure 6 continued*

correct and incorrect choices. Distribution of correlation coefficients (right). (**B**) Decodability of whether goal error occurred in upcoming or previous test phase based on population activity during delay period (mean and 95% CI). (**C**) Decodability of task phase from population activity (200 ms bins) while animal is moving inbound from goal to center in sample or test phase (pre-0 s) and after it arrives at center, for one animal. Similar results in another animal (not shown). (**D**)    t-distributed stochastic neighbor embedding (t-SNE) of population vectors of activity while animal is at key task locations: individual starts, goals, and bridges/routes.

The online version of this article includes the following figure supplement(s) for figure 6:

**Figure supplement 1.** t-SNE analysis for each animal and subarea.

unlikely to serve as general working memory correlates that can be employed under conditions of high behavioral flexibility, such as those often encountered in the real world. Whether animals in simpler tasks use such canonical representations of working memory to solve those tasks, employ an alternative strategy, or rely on an as-yet-undiscovered pattern of activity remains an open question. Our work therefore stresses the importance of employing more cognitively challenging tasks that allow dissociation between correlates of high-level cognitive variables and other task-related variables. For instance, the potential confounds resulting from such task-related variables are seen in the strikingly similar time course of goal decoding using either neural or behavioral tracking data outside of the well-controlled delay period (*Figure 3—figure supplement 4*). The search for a pure, flexible working memory correlate could focus both on other brain areas and on exploring as-yet-unobserved activity patterns or alternative memory mechanisms involving the mPFC. Such mechanisms could, for instance, be related to short-term synaptic plasticity (*Mongillo et al., 2008*), but differ from the previously reported covariance patterns (*Barbosa et al., 2020*) investigated here. Finally, our results suggest a role for mPFC in working memory tasks by representing task structure in terms of groups of behaviorally related elements (*Jung et al., 1998*; *Yu et al., 2018*; *Kaefer et al., 2020*), consistent with findings that the PFC forms long-term memories of learned stimulus categories (*Freedman et al., 2001*).

## Materials and methods

### Experimental procedures

#### Surgery

All procedures were conducted in accordance with the Janelia Research Campus Institutional Animal Care and Use Committee. The chronic Neuropixels implant surgery followed previously described methods (*Jun et al., 2017a*). Briefly, animals were anesthetized with isoflurane and mounted in a stereotaxic frame (Kopf Instruments). After thorough cleaning of the skull, a ground screw was placed through the skull above the cerebellum. A small craniotomy (diameter ~1 mm) was made above the target area (anterior-posterior: 3.24 mm, medio-lateral: 0.6 mm) in the right hemisphere. A single-shank Neuropixels '1.0' probe was lowered over the course of about an hour to a depth of 6.0–6.3 mm. The craniotomy was covered with artificial dura (Dow Corning Silicone gel 3–4680), and any parts of the probe outside of the brain were covered with sterile Vaseline. The probe was permanently fixed to the skull with dental acrylic, and a protective cone made of copper mesh and dental acrylic or light cured cement was built around the probe. Recordings started after animals had fully recovered and were accustomed to the recording cable when attached to the implant, ~2–3 weeks after implant surgery.

#### Behavioral procedures

Rats were housed in a reverse light cycle room (12 hr:12 hr day:night), and training and experiments were conducted mostly in the dark phase.

Rats learned the 'multi-start/multi-goal/multi-route' (MSMGMR) task over the course of several months in successive learning steps with generally one learning session per day. Animals were food restricted (with their weight maintained at ≥85% of their initial weight) to increase motivation to collect reward in our task.

Reward was given in the form of a sweet and nutritious liquid reward (Ensure Plus). The reward was dispensed from custom-made Teensy-operated reward pods which, along with the custom-

made nose ports and bridges (that provide available 'open' routes when up and are unavailable 'blocked' routes when down in this elevated maze environment), were controlled by custom-written finite-state machine software written in Matlab.

Data from three male Long-Evans rats were included in this study. (Two other animals were trained to high levels of performance for this study, but the recordings were lost before training was complete, in one case due to probe failure. In both of these cases, the probe was implanted before training began and the losses occurred after ~4 months of training. To limit such outcomes, the probe was implanted after training was complete in the subsequent animals [two of the three included ones]). Animals were ~12 weeks of age when training began for two animals and ~6 months for one animal.

To accustom animals to the elevated maze layout and the type of reward they would receive, animals were placed on the maze for ~30 min per session to explore and collect reward from any of the reward pods, which were located at the end of each goal arm and one in the center of the maze. All routes were available at this point in training and animals could freely move around to collect reward from any of these four reward sites on the maze (*Figure 1A*). Animals started exploring the maze immediately and learned to navigate between the reward sites within a few sessions.

Next, the sample phase of the task was introduced. Here, the animal learned the meaning of the visual cues (blue LEDs on the goal reward pod(s) blinking) and to find the current goal location by repeatedly being visually guided to the same goal and returning to the center after each run, which allowed it to understand the basic structure of the task (i.e. run out, run back to center, run out, and so on). This sample phase was implemented in two different configurations throughout the training and recording sessions for the three rats. For one rat, the correct goal location was indicated by a blinking light at the correct reward pod. For the other two rats, we used a reversed configuration where the correct goal was the only goal reward pod not blinking. This configuration was introduced to reduce the potential for the animals to remember the location of a simple visual cue instead of remembering the spatial location of the goal. However, both versions were readily learned by the animals and did not lead to any obvious changes in behavior in the test phase.

After the animal had learned to follow the guided cues, the test phase was introduced. Here the animal was cued three to four times to sample the same goal, followed by one run in which the animal was not cued and had to navigate to the same (correct) reward pod. If the animal went to an incorrect reward pod in the test phase first, they received a diminished or no reward if they then went to the correct one afterwards. The number of repeated 'sample phases' to the same goal was successively lowered until sample and test phases were interleaved.

Next, the use of a particular route was enforced during the sample phase. After animals returned from the test phase to the center reward pod, two of three bridges were lowered, that is the routes were blocked, forcing the animal to use the remaining one in the sample phase. The route available in each sample phase was chosen pseudorandomly.

Next, to introduce the nose pokes, all routes were blocked upon arrival of the rat at the center after the sample phase. The animal could choose any nose port; correct poking was indicated with a 4 kHz tone upon brief poking (50 ms). After holding for the specified duration, all routes would become available (i.e. all bridges were raised). At this stage, the duration of the required poke was successively increased from the initial 50 ms to approximately 1 s. Once the animal developed a habit of choosing the same nose port to make the routes available, only one pseudorandomly chosen nose port would elicit the tone and raise the bridges. The nosepoke duration was then successively increased until the animal became proficient at holding it for ~ 1.5 s. In two animals (the ones in which the goal was indicated as the pod that was not blinking), any incorrect nosepoke was indicated by a constant light at that port after it had been poked at least once.

As a final learning step, only one route would become available after the correct nose port poke (i.e. for the test phase) and the nosepoke duration was successively increased to 3 s (for two animals) and 3.2 s (for one animal).

The final version of the task used the following pseudorandom method for determining the goal, available sample phase route, correct nose poke, and available test phase route for each trial. The pseudorandom sequence of trials was determined anew for each session. There are 27 distinct combinations of start position, goal location, and test phase routes. To keep these combinations in balance overall and locally and to discourage formation of preferences for a particular goal, the order of these 27 combinations was randomly permuted with the constraints that (1) each of the three

subblocks of nine trials was also balanced to equally contain each start location, goal location, and test phase bridge and (2) the same goal was not repeated in more than two consecutive trials. The sample route only had the constraint that in a subblock of nine trials, for each goal location each of the routes was presented once in the sample phase. Identical blocks of 27 trials were repeated in a given session. Note that there was no indication given to the animal of the 27-trial block or 9-trial subblock structure, so the entire session appeared as a single long sequence of trials. In this final version, the animal was not allowed to correct an error in the test phase and had to go to the center to initiate a new trial. In addition to the given pseudorandomly determined sequence of trials, if the animal made an error in the test phase of a trial, that same trial could be repeated one time (but with a potentially different sample route). In terms of discouraging preferences for a given goal, repeating a trial due to an error would necessarily mean the animal was not repeating a trip to the same goal in the test phase.

After the animal was able to perform the full task, training was continued until the animals reached ~70% accuracy (total training time from naïve animal to this point was ~3.5–6 months). Then (for two animals), a Neuropixels probe was implanted in the mPFC (while for one animal, the Neuropixels probe had been implanted before training). Recording began after the animal recovered and was acclimated to the recording cable.

Between training sessions, the maze was wiped with 70% ethanol to reduce any odor cues animals might use to navigate to the correct goal. Furthermore, the maze was mounted on a turntable-like frame and rotated to one of three orientations in between sessions, to further lower the probability that animals used local cues to remember goal locations. The maze was set up in a room with multiple cues outside of the maze, such as other lab equipment. Care was taken to ensure that the reward pods at the goals all delivered the same amount of reward and appeared visually identical. These precautions were taken to ensure that animals learned to use distal, non-local cues for navigation and to encode the currently rewarded goal location.

After conclusion of the experiments in the standard maze configuration, we performed a behavioral manipulation experiment in two of the three animals in which the maze was rotated by 60 degrees, so that the reference frame changed and the goal and start positions were in between their previous positions. The initial drop in performance we observed after the rotation of the maze in the relearning experiments (*Figure 4*) is in line with animals using distal, non-local cues for navigation.

## Histology
After conclusion of the experiments, animals were deeply anesthetized and underwent transcardial perfusion with saline followed by 4% paraformaldehyde for fixation. Brains were removed and sectioned for histological verification of the recording site. Location of PFC subareas was estimated based on the entry point of the probe into the brain (after the section had been aligned to the corresponding one in the atlas) and implant depth.

## Electrophysiology
Neural data from Neuropixels '1.0' probes (https://www.neuropixels.org) was recorded with SpikeGLX software (http://billkarsh.github.io/SpikeGLX/). Three hundred and eighty four channels were recorded simultaneously across subareas of the mPFC in two separate frequency bands (spike: 300 Hz to 10 kHz sampled at 30 kHz and LFP: 0.5–300 Hz sampled at 2.5 kHz). The recording system and a laptop capturing the digitized data from the probe were mounted on a manually controlled, motorized rotating platform mounted at the ceiling to avoid the cable from becoming too twisted from the animals' turning. This apparatus was used for two of three animals. For one animal, the experiment was briefly interrupted to 'untwist' the cable by rotating the animal when it became necessary.

## Data analysis
All data analysis was done using custom-written programs in Matlab or Python, and for some machine learning procedures, the scikit-learn library was used (*Pedregosa et al., 2011*).

## Behavioral data

Animal head location and orientation during neural recording were tracked at 30 Hz with two differently colored LEDs attached to the implant. To assess vicarious trial and error behavior, the speed of the animal (across windows of 330 ms) was analyzed in the test phase. Events were counted as candidate vicarious trial and error behavioral events when the speed of the animal dropped (for at least one window) below 5 cm/s in the 800 ms time window centered around the choice point (when the animal has crossed the bridge and enters the outer ring). We found that in 8, 3, and 4 trials, the speed dropped below the threshold in this manner for each of the animals, out of 62, 110, and 64 total correct trials, respectively.

Unless otherwise noted, only correct test phase trials were used for analysis. After removing the trials in the beginning as described below, the numbers of total trials (numbers of correct trials) per animal were 79 (62), 81 (64), and 145 (110).

## Preprocessing of neural data

Multiple sessions were recorded from each animal in the standard maze configuration, but only one was included in the analysis per animal as the probes were not moveable and the population of cells could not be assumed to be independent across different recording sessions. The session to be included in the analysis from each animal was selected based on a combination of good behavioral performance and a high number of trials. For the behavioral manipulation presented in *Figure 4* (rotation experiment), two animals that were previously trained and recorded in the standard configuration underwent this experiment. Because this experiment required novelty, only the first session in which the manipulation was conducted was used for analysis. JRCLUST (*Jun et al., 2017b*) (version 08/2019, Vidrio) was used to automatically presort the spike data and then manually curate it afterwards. To allow the animal to settle into the behavioral task and to remove global drifts leading to changes in firing rate across a significant number of cells (which were observed to occur at the beginning of each session, presumably due to the handling of the rat necessary to attach the probe to the cable), 3–20 trials were removed from the beginning of each session. Because we were searching for a working memory code that was stable and robust throughout the session, and to reduce the possibility that non-stationarities would reduce the performance of the decoders, we selected for analysis the subset of cells that satisfied the following three stability/robustness criteria applied to each cell separately (but we also performed the main analyses including all cells without such selection, *Figure 3—figure supplement 3*). First, the overall firing rate had to be stable across the session: specifically, a linear regression on the standardized firing rate in 10 s bins over the session was performed and the absolute difference between the first bin and the last bin could not exceed 1 (i.e. the slope of any change in firing rate needed to be within ±1 s.d./n, where n = the total number of 10 s bins). Second, the firing rate in delay periods had to be stable across the session: specifically, the absolute difference of a linear regression on the summed spike count for all delay periods (3 or 3.2 s each) between the first and the last delay period could not exceed 1.4 (note that the different goals were pseudorandomly distributed across a session, so that cells selective for only one goal would not be excluded this way). Third, the cell needed to be active in a minimum number of delay periods: specifically, the cell had to fire at least one spike in at least one-sixth of all delay periods (set to potentially allow for a cell that was active in half of the delay periods for a particular goal and silent otherwise). The numbers of clusters isolated during spike sorting were 182, 131, and 98 for the three animals (152 and 186 for the relearning sessions) and, after applying the criteria above, 103, 86, and 72 (88 and 111 for the relearning sessions) cells remained. Putative fast-spiking GABAergic interneurons were excluded from analysis based on having a combination of faster waveform (shorter peak to trough interval) and higher firing rate across the whole recording duration, resulting in 97, 84, and 68 active, stable, putative principal cells for the three animals (84 and 105 for the relearning sessions).

## Correlation analysis

For the correlation matrices in *Figure 2A,B*, a 'goal arrival' PV for each trial was calculated from the 3 s period after the animal had arrived at the goal in the sample phase (where time point 0 was assigned to be the time that the infrared beam on the reward pod was broken, which triggers delivery of the reward). Similarly, for all test phases, a delay period PV was calculated. The matrices

containing the raw firing rates were concatenated, and the Pearson's correlation coefficient was calculated for all combinations of PVs. In *Figure 2A*, only the correlations among goal PVs are shown, and in *Figure 2B*, the correlation between delay period PVs and goal PVs is shown.

To test whether the remembered goal is preferentially represented in the delay period over time (*Figure 2C*), we calculated the average PV from all sample phases when the animal was at one specific goal and correlated the resulting three PVs with each time bin in all delay periods. For each delay period, the 'winning' goal was either the one with the highest mean correlation with that goal across all bins or the one that had the highest number of time bins in which the correlation was highest with that goal (majority vote). A corresponding approach was taken to classify each delay period with regards to the start (current location), except the current delay period was excluded from the average of the start PV. Here and elsewhere, a bootstrap analysis was used to calculate the 95% CI of the decoding accuracy: for each binwidth, 1000 - 10,000 samples were drawn randomly from all trials from all animals with replacement and the 2.5 and 97.5 percentile values of the means were taken as the interval.

## H-score analysis

To assess what information is encoded in the delay period at the single-cell level (*Figure 3—figure supplement 2*), the spikes in each delay period were binned using a variety of binwidths and each cell × bin was considered one sample of a class. Several types of classes were considered separately: the current start (current location), the remembered goal, the goal in egocentric coordinates (i.e. behind the start, to the left, or to the right), or the combination of the start and the remembered goal (3 × 3 classes). The distributions of spike counts of samples belonging to different classes (e.g. the different starts) were compared using the Kruskal–Wallis test. To correct for multiple comparisons, false discovery rate correction was applied to each binwidth tested.

## Supervised machine learning classification methods

Generally, for each classification method, a range of hyperparameters were tested and a set of parameters that reached the highest cross-validation accuracy for start (i.e. current location) decoding was chosen for each method and kept constant. A leave-one-out cross-validation scheme was used for all classification methods. The numbers of samples per class were balanced throughout by randomly subsampling from the class(es) with the higher number of samples in the training set. Decoding accuracy was reported as the mean of the cross-validated accuracy over all classes. For population analyses where the delay period was binned in time (*Figure 3B,C*, *Figure 2—figure supplement 1B*, *Figure 3—figure supplement 1*, *Figures 4B* and *6B,C*), all bins of a given delay period were concatenated into one vector and each cell × bin was treated as a separate feature, that is the activity patterns considered were fixed with respect to the starts of the delay periods (and analogously for the phase bins in *Figure 5A*). The matrix size used for classification was thus # of time bins times multiplied by # of cells x # of trials. Each feature was standardized over all trials by subtracting the mean and dividing by the standard deviation (std), unless stated otherwise. Logistic regression (*Figures 2A*, right, 3C, *Figure 3—figure supplements 3* and *4*, *Figures 4* and *6B*) was used with L2 regularization. For the time-resolved decoding in *Figure 3—figure supplement 4*, we used overlapping windows of 800 ms (200 ms bins, steps of 200 ms) for the neural data and 330 ms for the position tracking data in (steps of 100 ms).

The support vector machine classifier (*Figure 2—figure supplements 1B*, Figure 3C, *Figure 3—figure supplement 3* and *Figure 6C*) was used with a Gaussian kernel to allow for non-linear decision boundaries. The kernel coefficient was set to 0.001, and L2 regularization was used. In *Figure 3B*, left bottom, the data was divided into nine classes, corresponding to the nine possible combinations of remembered goals and current start locations. For *Figure 3C*, three classes corresponding to either the three possible start locations, three allocentric goals, or three egocentrically defined goals were used. Correspondingly, for the analysis of task phase and spatial selectivity at the goal (*Figure 2—figure supplement 1*; note the time period analyzed was the 3 s after the animal had arrived at the goal as in *Figure 2*), the data were divided into six classes corresponding to the six possible combinations of goal location and task phase. For the summary plot containing all animals in *Figure 2—figure supplement 1*, middle, only the classification accuracy of either phase or goal was considered. For *Figure 2—figure supplement 1* right, two classifiers were trained, one

with all data from the sample phase and one with all data from test phase. The remaining data were each used to predict the goal it encoded, that is data from the test phase were fed into the classifier built from sample phase data and the other way around. Decoding accuracy was given as the mean over all three classes. In *Figure 6C* for trial phase decoding at the center, we only compared trials in which the animal took the same bridge back to center, so that direction of arrival at the center was comparable.

For the random forest classification in *Figure 3C*, *Figure 3—figure supplement 3*, the data was prepared and balanced as described above, and the forest contained 1000 trees for each classifier. The maximum number of features considered for finding the best split was chosen to be $\sqrt{n}$, with $n$ being the number of features considered, that is for smaller time bins where the number of features is higher as described above, more features would be considered for each split.

For the Naïve Bayes classification (*Figure 3C*, *Figure 3—figure supplement 3*), Gaussian distribution of the features was assumed, and for each classifier, only the 10% of features with the highest H-scores (from Kruskal–Wallis test) were used.

## LFP-phase analysis (*Figure 5A*, *Figure 5—figure supplement 1*)

LFP channels that corresponded to references or were noisy (std, either lower than 1/4 of the mean std or four times higher than the mean std) were removed. The LFP trace considered for a given cluster was the average of 10 LFP channels that were at least eight sites away in both directions from the center of the cluster (i.e. the site with maximum amplitude) whose phase was analyzed, generally consisting of five sites above and below the center of the site (but if the center was too close to the edge of a block of recorded channels, the 10 channels used to average could be split unequally, e.g. eight sites above and two sites below for a cluster near the bottom of a block). The LFP from 3 s before to 3 s after the delay period was filtered in the 2.5 to 5, 5 to 12, or 15 to 30 Hz (FIR filter), but only the delay period itself was considered for phase analysis. The phase of the oscillation of a frequency band was determined by calculating the angle of the Hilbert transform. Periods in which the resulting phase was not monotonically increasing between peaks were rejected (mean time rejected per delay period, across animals and the three frequency bands: 194 ms, maximum time rejected: 1.52 s, out of 3–3.2 s total) and spike times were mapped onto phases. For each cell, the spike phases from all delay periods were divided into three classes, corresponding to the currently remembered goal. The length of the mean phase (resultant vector length) was computed as a measure of preferred firing phase for each cell and class. To test whether firing phase contained any information about the remembered goal location, the labels (remembered goal 1, remembered goal 2, or remembered goal 3) of delay periods were shuffled and the distribution of resultant vector lengths were compared to the one from the actual labels (*Figure 5A*, middle). In *Figure 5—figure supplement 1C*, the same approach was taken, but only cells that showed significant phase locking over all correct trials were used. In a separate approach (*Figure 5A*, right, *Figure 5—figure supplement 1C*), the phases of all spikes were binned into 2, 6, or 12 bins, corresponding to the number of spikes that were elicited in a particular phase bin (e.g. one of the bins for the 2-bin case included phases from −90 to 90 degree). All phase bins for each cell were concatenated and used as features for a logistic regression classifier trained on all but one test trial and tested on that trial (i.e. leave-one-out). *Figure 5—figure supplement 1C* only cells that showed significant phase locking to at least one of the goals were used (only the training set was used for selecting the phase-locked cells). To account for differences in total valid duration of each delay period (which could be less than the full duration due to the existence of periods with non-monotonically increasing phase), the counts in each bin in each delay period were divided by the total valid duration of that delay period. To account for differences in overall spike rate, these adjusted counts were normalized by subtracting the mean and dividing by the std over all delay periods for a given feature.

## Covariance analysis (*Figure 5B*, *Figure 5—figure supplement 2*)

Covariance was analyzed using the method described in *Barbosa et al., 2020* and parts of the associated code at (https://github.com/comptelab/interplayPFC) was used in modified form. The following adaptations were made to fit our experimental data. Only cells that were significantly modulated at the sample goal location (i.e. different for different goals) were included in the analysis. A cell's

'preferred goal location' was the one where it had the highest firing rate. Only pairs of neurons that shared the same preferred goal location were considered for analysis.

As in *Barbosa et al., 2020*, spikes for all trials were binned in 10 ms bins and shuffled in steps of 50 ms. The cross-covariance was calculated for each shuffle (1000) and the mean subtracted from each trial to remove any dynamics faster than 50 ms. The resulting (jitter-corrected) cross-covariance was taken to be the mean of the three bins around the 0-lag bin. In the case of the single time point analysis, the full delay period was considered (3 or 3.2 s). For the time-resolved version in Figure 5-figure supplement 2 bottom, time windows of 1 s were used and cross-covariance was repeatedly calculated in steps of 50 ms. An 'excitatory pair' of neurons was considered as such if the sign of the mean jitter-corrected covariance was positive both for the preferred and non-preferred trials and conversely considered an 'inhibitory pair' if the sign was negative for both. Pairs with inconsistent signs were discarded. The sign was calculated separately for the delay period only and the extended delay period in Figure 5-figure supplement 2 for the time-resolved version (−2–5 s, with 0 being the beginning of the delay period).

This procedure resulted in the following numbers of total pairs/excitatory/inhibitory for the full delay period: rat 1: 288/67/67, rat 2: 339/88/82, rat 3: 232/61/59.

For the time-resolved version, these numbers were as follows: rat 1: 288/72/59, rat 2: 339/72/84, rat 3: 232/62/66.

The cross-correlation selectivity index (CCSI) (*Barbosa et al., 2020*) for the excitatory pairs was the mean difference of the cross-covariance in preferred and non-preferred trials, and similarly for the inhibitory pairs. The numbers of preferred and non-preferred trials were matched (by randomly subsampling the non-preferred trials).

## t-distributed stochastic neighbor embedding – analysis (*Figure 6C*, *Figure 6—figure supplement 1*)

For t-distributed stochastic neighbor embedding (t-SNE), firing rate PVs from all delay periods (correct trials), the period 0–3 s after goal arrival in the sample phase, and activity during crossing of the bridge after the delay period (crossing time: 0.28 s on average) in the test phase were embedded in two-dimensional space according to *Maaten and Hinton, 2008*; perplexity was set to 35 and the learning rate to 100. Ten embeddings were calculated for each data set, and the embedding with the lowest Kullback-Leibler divergence between data and embedding is shown. The overall structure of embeddings was stable over multiple runs and a range of perplexities and learning rates. Hyperparameters were kept constant when the embeddings were calculated separately for subareas.

## Acknowledgements

This work was funded by the Howard Hughes Medical Institute. We thank S Erwin, R Gattoni, P Rich, J Jun, B Karsh, J Colonell, B Barbarits, W Sun, T Harris, J Chen, J Arnold, S Sawtelle, P Polidoro, Vidrio, S Romani, and K Branson for assistance and advice. We thank A Hermundstad, A Hantman, S Romani, W Asaad, A Dorrn, and J Dudman for comments on the manuscript.

## Additional information

### Funding

| Funder | Author |
| --- | --- |
| Howard Hughes Medical Institute | Claudia Böhm |
|  | Albert K Lee |

The funders had no role in study design, data collection and interpretation, or the decision to submit the work for publication.

### Author contributions

Claudia Böhm, Conceptualization, Visualization, Writing - original draft, Writing - review and editing, performed experiments and analyzed data; Albert K Lee, Conceptualization, Supervision, Funding acquisition, Writing - review and editing

**Author ORCIDs**
Claudia Böhm (iD) https://orcid.org/0000-0002-9802-1162
Albert K Lee (iD) https://orcid.org/0000-0003-4332-8332

## Ethics

Animal experimentation: All procedures were conducted in accordance with the Janelia Research Campus Institutional Animal Care and Use Committee (permit number #17-158).

## Decision letter and Author response

Decision letter https://doi.org/10.7554/eLife.63035.sa1
Author response https://doi.org/10.7554/eLife.63035.sa2

## Additional files

### Supplementary files
• Transparent reporting form

## Data availability

The data that support the main findings of this study are available at https://github.com/LeeA-Lab/Boehm_Lee_MSMGMR (copy archived at https://archive.softwareheritage.org/swh:1:rev:55d28ccf0459c33e6009d4cd66edb37e9b7870c4/).

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
