## [Decision Letter]

**Acceptance summary:**

The ability to maintain information relative to a goal is crucial for successful behaviour and is believed to depend on the prefrontal cortex in mammals. This papers combines cutting edge techniques to record large ensembles of prefrontal neurons in freely moving rats performing goal-oriented behaviour and advanced data analysis methods. No neuronal correlates of the goals were found in the prefrontal cortex, thus possibly invalidating a large class of computational models.

**Decision letter after peer review:**

Thank you for submitting your article "Canonical goal representations are absent from prefrontal cortex in a spatial working memory task requiring flexibility" for consideration by *eLife*. Your article has been reviewed by three peer reviewers, and the evaluation has been overseen by a Reviewing Editor and Laura Colgin as the Senior Editor. The following individual involved in review of your submission has agreed to reveal their identity: Vincent Hok (Reviewer #3).

The reviewers have discussed the reviews with one another and the Reviewing Editor has drafted this decision to help you prepare a revised submission.

This paper reports how prefrontal (PFC) population activity does not obviously maintain trial-relevant information. Specifically, by recording from populations of neurons in freely moving rats during a spatial working memory task, the authors did not find any neuronal correlates of memory in the interval between sampling and choosing a rewarded goal location. While the three reviewers are enthusiastic about the findings, they have all stressed out the importance of improving controls by further analyzing the reported data and adding control data. If recordings were performed in these animals during the retraining phase following surgery, the reviewers have agreed that analyzing these data would certainly be sufficient to support the main claims of the study. You will find below a point-by-point summary of the major concerns raised by the reviewers. Please refer to their reviews for more details .

1) One potential confound of the present data is that the animals perform flexible route planning and maintain a memory of the goal position simultaneously. The authors did not separate the two aspects of the task. While, ideally, the authors would have additionally performed the task with fixed routes to disambiguate the contribution of route planning and spatial working memory, the authors should analyze data acquired during retraining after surgery – if these data were collected. A (likely) lower performance during retraining would allow the authors to further investigate how neuronal activity correlates (or not) with task parameters.

2) At any rate, the present data should be more carefully analyzed. First, along the lines of the last comment, the authors should report any difference between correct and error trials.

3) Furthermore, working memory during this task may be loaded/encoded as soon as the rat observes the sample cue light, which indicates a given trial's goal location. For this reason, it is reasonable to assume that the "delay period" starts and ends considerably beyond the 3s nose poke. Analyses extending to a fuller task timeline are required to provide a full picture of the neuronal processes at play during the task. See for example the study by Maggi et al., 2018, where the authors showed that neuronal correlates of task parameters appeared as the animals return to the starting point.

4) Different PFC neurons likely show different correlates to task parameters. The authors should provide statistics relative to the proportion of units showing goal location or any other representation of interest and investigate whether these representations are captured by the population-level analysis.

5) Can the authors be sure the rats can distinguish the reward magnitude, in other words discriminate between sample and test phases? Are there distinct neural correlates of these features?

6) The authors should report more details regarding animal behaviour. For instance, how often do animals progress directly to the correct goal, rather than trying each option sequentially? Do they always take the shortest path? If not, how does search behaviour progress through training? Does making an initial error lead to a faster run on the final approach to the correct goal? Do they show vicarious trial and error behavior in the task?

7) The LFP analysis should be improved. It may be that only a subset of the PFC population significantly phase-lock to 6-12Hz or 15-30Hz oscillations, but this would be missed by the current treatment. If 6-12Hz and 15-30Hz were chosen on the basis of "power during the delay", how did power vary? What about γ rhythms (see γ burst during working memory by Earl Miller and colleagues)?

Reviewer #1:

I enjoyed reading this cognitive neuroscience manuscript, which combines a novel assay of goal-directed behavior in rat with chronic electrophysiological recording of prefrontal (PFC) cortical network activity. The experiments are well-designed and clearly described, the manuscript is readable and succinct, and analyses are sensibly motivated by the literature. The emphasis lies on a negative finding: that PFC population activity does not obviously maintain trial-relevant information during a stage of the task that occurs between sampling and choosing a rewarded goal location.

One key challenge for the authors is therefore to rule-out a false negative. This is not to say that a representation absolutely must be in the PFC somewhere and at some point during this task. After all, the task itself is novel and training extended over several months, making direct comparisons with existing data from (e.g.) T-mazes difficult. It is conceivable – but unlikely – that rats could perform the task without a functional PFC. The ideal would be to see the same rats perform a "canonical" working memory task, and check whether "canonical" representations do emerge under "canonical" conditions. A varied delay length would also be of interest; but I accept neither of these is a reasonable ask during a pandemic. With this in mind, I have the following comments:

1) The title is unintentionally misleading, since it states that representations are absent – but they have only been hunted during the 3s nose-poke / delay stage of the task.

2) In principle, working memory during this task may be loaded/encoded as soon as the rat observes the sample cue light, which indicates a given trial's goal location. So the "delay period" starts and ends considerably beyond the 3s nose poke. I think equivalent analyses extending to a fuller task timeline are required to make (more) sense of what is going on.

As an example, Maggi et al., 2018, analyzed rat PFC activity on a Y-maze. They show ensemble encoding of goal as rats return to the centre of the maze post-reward, but only during recording sessions associated with an inflexion in behavioral performance. Considering more task stages and relating analyses to behavior beyond picking a "good" session for each rat may change the picture.

3) Most analyses treat the PFC population as one. What proportion of units signal goal location? What if a subset of neurons carry the representation(s) of interest? Would this be captured by the current approach?

4) Can the authors be sure the rats can distinguish the reward magnitude, in other words discriminate between sample and test phases? Are there distinct neural correlates of these features?

5) The LFP phase analysis is briefly described and quite cursory. For instance, it may be that only a subset of the PFC population significantly phase-lock to 6-12Hz or 15-30Hz oscillations, but this would be missed by the current treatment. If 6-12Hz and 15-30Hz were chosen on the basis of "power during the delay", how did power vary? What about γ rhythms (see γ burst during working memory by Earl Miller and colleagues)?

6) What happens on error trials?

7) Description of the animals' behavior is brief and requires further important details of this novel task in order for readers to grasp how rats might arrive at potential solutions. For instance, how often do animals progress directly to the correct goal, rather than trying each option sequentially? Do they always take the shortest path? If not, how does search behaviour progress through training? Does making an initial error lead to a faster run on the final approach to the correct goal? Do they show vicarious trial and error behavior in the task?

Reviewer #2:

In this manuscript, Bohm and Lee investigated the neuronal representations in the PFC during the spatial working memory task that needed flexible navigation. In this task, the rats were required to remember the goal position, and the routes from the start to the goal varied every trial. They recorded neuronal activities in the PFC during the task and found that PFC neurons did not represent goal-positon-related information during the delay periods at the start position. The topic is of great interest, and the paper contains some intriguing results. However, there is one major concern that should be addressed to support their claims.

1) One crucial issue is that the paper is missing proper control experiments. Because the authors attempt to claim that "PFC neurons represents goal-related information in the working memory task without flexible route navigation (e.g., the routes are fixed), but do not represent it in the working memory task with flexible route navigation," they need to perform both the route-flexible and route-fixed working memory tasks with the same experimental apparatus. Otherwise, we cannot assess whether the lack of differential activity patterns of PFC neurons was due to the task structure (i.e., due to flexible navigation) or other reasons.

Reviewer #3:

First of all, I would like to thank the authors for the great clarity of their manuscript, the quality of the analyses and the associated figures. Although it is difficult for me to rule on the relevance of certain parameters used in the t-SNE analysis, I was overall more than convinced by the results produced. In fact, the authors, in my opinion, performed all the relevant analyses to try to extract as much information as possible from their data set.

Nevertheless, one major question remains to be addressed. Given the extremely long learning time in this type of task, one may wonder whether the prefrontal cortex remains involved.

1) Because of the large amount of work this would represent, it would be unreasonable to ask the authors to perform a control experiment aiming at inactivating the prefrontal cortex when the animals master the task. Perhaps an acceptable solution would be to conduct the same analyses when the animals reacquire the task post-surgery, or at least when their overall performance level is significantly lower than that presented in the paper and significantly higher than the level of chance. The idea would be that in this reacquisition phase, the prefrontal cortex would be more likely to show correlates related to route planning.

2) The authors mention that one of the three animals was implanted prior to task acquisition. Although it is difficult to draw definitive conclusions from a single animal, perhaps it is possible to analyze the activity of the prefrontal cortex in this animal during the learning of the last behavioural phase.

---

## [Author Response]

This paper reports how prefrontal (PFC) population activity does not obviously maintain trial-relevant information. Specifically, by recording from populations of neurons in freely moving rats during a spatial working memory task, the authors did not find any neuronal correlates of memory in the interval between sampling and choosing a rewarded goal location. While the three reviewers are enthusiastic about the findings, they have all stressed out the importance of improving controls by further analyzing the reported data and adding control data. If recordings were performed in these animals during the retraining phase following surgery, the reviewers have agreed that analyzing these data would certainly be sufficient to support the main claims of the study. You will find below a point-by-point summary of the major concerns raised by the reviewers. Please refer to their reviews for more details.1) One potential confound of the present data is that the animals perform flexible route planning and maintain a memory of the goal position simultaneously. The authors did not separate the two aspects of the task. While, ideally, the authors would have additionally performed the task with fixed routes to disambiguate the contribution of route planning and spatial working memory, the authors should analyze data acquired during retraining after surgery – if these data were collected. A (likely) lower performance during retraining would allow the authors to further investigate how neuronal activity correlates (or not) with task parameters.

We appreciate this valuable suggestion. One animal received surgery before training but was not recorded in the training phase. In the other two animals, we initially retrained the animal using a dummy cable to habituate them to running with the implant and cable. We did collect some data during retraining, but the animal’s performance could be due to residual habituation to the cable, any residual recovery, etc, as well as the cognitive demands posed by relearning. However, we did perform a behavioral manipulation experiment in two of the three animals after they were fully retrained (after the recording sessions analyzed in the original manuscript), which allowed us to analyze activity during learning/relearning in this task with high trial numbers and free of potential confounds. In these experiments we rotated the maze, which is mounted on a rotatable frame, by 60 degrees so that the reference frame changed and the goal and start positions were in between their previous positions. Animals had never seen this configuration before or witnessed a rotation of the maze. We conducted the rotation approximately a third of the way through the session. Consistent with the idea that the animals must adapt to the new configuration and relearn to apply previously internalized rules, their performance dropped dramatically and gradually improved over the course of the session. We originally conducted these experiments to study remapping of a distributed spatial code such as the one we observed in prefrontal cortex.

However, since these appear to be ideal data sets to answer the question posed by the reviewer about the effect of learning on the representation of task variables, we have included them in this manuscript. We were able to sort 152 (84 stable pyramidal cells) and 186 (105 stable pyramidal cells) mPFC neurons from each animal during performance in rotation sessions that consisted of 45 and 40 trials in the standard configuration followed by 104 and 88 trials after rotation, where the number of trials performed after the manipulation is as many as in the standard sessions included in the original manuscript. We tested whether the higher cognitive demand during relearning could elicit a detectable representation of the currently remembered goal during the nose poke fixation delay period. Applying the various population-level machine learning classification methods at multiple time resolutions as in the standard sessions, we found no evidence for such representations. These data provide further evidence that flexible working memory is not encoded in prefrontal cortex in a canonical form. In contrast, spatial representation of the start locations is again strongly represented even as the animal adapts to the new configuration. We have added these two new data sets and analysis to the revised manuscript, in a new Figure 4, and in the Results, Discussion, and Materials and methods sections.

2) At any rate, the present data should be more carefully analyzed. First, along the lines of the last comment, the authors should report any difference between correct and error trials.

We have now extended our analysis of correct and error trials beyond the delay period (nose poke). First, we have analyzed the population vector activity at the goal during the sample phase, when the current goal is presumably encoded for later use in the test phase of the trial. The encoding during the sample phase is of particular interest in our task design because the current goal can change trial by trial as our task does not have a block structure. Inspired by the work of Maggi et al., 2018 (which was pointed out by the reviewers in point #3), we tested whether the correlation between the population vectors of prospective correct trials differed from the correlation between the population vectors between prospective error trials, and we found no difference. Second, again inspired by Maggi et al., 2018, we have analyzed data retrospective to the outcome of a trial. Here we found that activity after the animal has made an error is markedly different from activity in correct trials. This could be explained by the presence of reward in correct trials and the missing reward in error trials or the resulting differences in behavior. However, correlations between correct trial population vectors were similar to correlations between error trial population vectors. These analyses complement our survey of error information in the delay period in the original manuscript. We have included these new analyses in new panels to Figure 6 and have added the following text to the Results section:

“First, we analyzed if the activity at the goal in the sample phase (presumably during encoding) differs when there is an error in the subsequent test phase or not, but this was not the case. In contrast, after the animal had made an incorrect choice in the test phase, activity at the goal was markedly different, presumably due to the lack of reward; however, correlations between the population vectors of activity at the goal among correct trials and, separately, among error trials was comparable (Figure 6A)”

3) Furthermore, working memory during this task may be loaded/encoded as soon as the rat observes the sample cue light, which indicates a given trial's goal location. For this reason, it is reasonable to assume that the "delay period" starts and ends considerably beyond the 3s nose poke. Analyses extending to a fuller task timeline are required to provide a full picture of the neuronal processes at play during the task. See for example the study by Maggi et al., 2018, where the authors showed that neuronal correlates of task parameters appeared as the animals return to the starting point.

We fully agree that an analysis of other time periods than the controlled delay period (nose poke) are of great interest and we have conducted a comprehensive survey throughout the task timeline. We have analyzed our data in a temporally resolved fashion aligned to multiple reference time points during trial progression. Specifically, we tested how well the current goal can be decoded surrounding the time points: (1) when the animal arrives at the goal during the sample phase, (2) when the animal returns to the center during the transition between sample and test phase, (3) when the route becomes available (the end of the nose poke delay period), (4) around the choice point when the animal enters the outer ring in the test phase, and (5) when the animal arrives at the goal during the test phase. Importantly, we have conducted these analyses not only for the neural data but also for position tracking data that we obtained by following the two LEDs attached to the animals’ head during recordings. These data show that it is possible to decode the currently remembered goal location from the neural data at various other time points. However, a comparison to the ability to decode the current goal location based on position tracking data reveals a very high similarity in the time course of the appearance of a current goal signal in the neural data and the tracking data. Thus, it is likely that the apparent representation of the currently remembered goal is likely a representation of the animal’s position, posture or other behavioral features that happen to be correlated with the location of the current goal.

These data emphasize the importance of and validate our novel task design, which is specifically designed to feature a specific delay period (the nose poke fixation delay) during which any representation, if found, is free of other behavioral correlates. These data also serve as an important control as they show that the lack of a representation in the delay period (nose poke) is not due to insufficient amounts of data. We have included these new analyses as Figure 3—figure supplement 4.

4) Different PFC neurons likely show different correlates to task parameters. The authors should provide statistics relative to the proportion of units showing goal location or any other representation of interest and investigate whether these representations are captured by the population-level analysis.

To address this point we analyzed the spatial selectivity for start (i.e. nose poke delay) locations and goal locations for each animal at the single-cell level (the analysis for remembered goal locations is included in the manuscript in Figure 3—figure supplement 2). For each neuron we tested if its firing rate is significantly different for start locations (when the animal was at those locations during the nose poke delay) or goal locations (when the animal was at the goal itself in the sample phase). We have now included displays of the proportions of cells that are selective for start only, for goal only, for both or for neither. Either type of selectivity was found in all animals: the proportion of neurons that showed any spatial selectivity ranged between 54 and 72 %. Between 25 and 29 % of neurons distinguished only between goal locations and 9 to 16 % only between start locations, while 15 to 31 % of neurons had significantly different firing rates both at different goals and at different starts. To further investigate if cells of a given selectivity were predominantly found in a particular subarea, we investigated their distribution across subareas. We found that cells with any type of selectivity could reside in any subarea.

For population-level analysis, especially correlations, it is conceivable that goal-location specific activity in the delay period is concealed by neurons that do not contain goal-location specific activity. We thus searched for goal-location specific population activity in the delay period including only neurons that show goal-location specific activity at the goals themselves. The result strongly resembles the one found when including all neurons, again suggesting that current goal information is not maintained in canonical form in the delay period, even when measures like this to improve the signal to noise ratio are taken.

We have included these new analyses in a new figure supplement (Figure 3—figure supplement 1) and present these new findings in the Results section.

5) Can the authors be sure the rats can distinguish the reward magnitude, in other words discriminate between sample and test phases? Are there distinct neural correlates of these features?

We analyzed the data while the animal is at the goal and compared the activity in the sample and in the test phase (which is shown in Figure 2—figure supplement 1B). Indeed, we can not only decode which goal the animal is currently at but also almost perfectly if the animal is visiting the goal in the test or in the sample phase. This might reflect the different amount of reward the animal receives or the different task phases per se. When animals return to the center they receive a small amount of reward which is not different in test and sample phases. Thus, we also tested if we can decode task phase in this period (Figure 6C). This was only possible after there were other sensory cues indicating the task phase (such as the lights going on) or the behavior was markedly different – suggesting that “pure” task phase might not be represented. We have expanded and clarified the description of these findings in the result section of the revised manuscript.

6) The authors should report more details regarding animal behaviour. For instance, how often do animals progress directly to the correct goal, rather than trying each option sequentially? Do they always take the shortest path? If not, how does search behaviour progress through training? Does making an initial error lead to a faster run on the final approach to the correct goal? Do they show vicarious trial and error behavior in the task?

To avoid confusing the animal, we did not allow the animal to correct themselves. If the first choice was wrong, they had to return to center and would be presented with a new sample phase. Thus, none of the trials which are counted as correct are preceded by incorrect choices.

Regarding the shortest path: In the test phase of trials, in the ~2/3 of cases in which one of the bridges adjacent to the current goal location is the one that is made available, rats most often take the shortest route (in 88%, 81% and 95% of trials for each of the animals, respectively), suggesting the animal indeed uses a map to navigate instead of a route planning or recognition strategy. We have added this information to the manuscript in the Results section. (Note that in the ~1/3 of cases in which the available bridge/route is opposite to the current goal, making a turn in either direction leads to routes to the goal of equal length.)

Regarding how behavior progresses through training: Animals are trained in stages, thus the strategy the animal chooses is impacted by the experimental design. Only early on in the training are the animals cued to go to the same goal more than one time. This only serves to teach the animal to understand the cues. As soon as the animal understands how the cueing works (which takes only a few sessions), each goal is only cued once in the sample phase, followed by the test phase. If the animal made an error in the sample phase (this is very rare), we repeat the sample phase to make sure the animal knows where the current goal is.

Regarding vicarious trial and error behavior: We have analyzed the speed of the animal (in windows of 330 ms length) in the test phase and find that in the majority of trials the animal did not stop to consider its choices. Specifically, we counted events as candidate vicarious trial and error behavior when the speed of the animal dropped below 5 cm/s in the 800 ms time window around the choice point (when the animal enters the outer ring after it crosses a bridge). We found that in 8, 3 and 4 trials the speed dropped below the threshold in this manner for each animal, out of 62, 110, and 64 total correct trials, respectively. We have added this observation to the Results and Materials and methods section of our revised manuscript.

7) The LFP analysis should be improved. It may be that only a subset of the PFC population significantly phase-lock to 6-12Hz or 15-30Hz oscillations, but this would be missed by the current treatment. If 6-12Hz and 15-30Hz were chosen on the basis of "power during the delay", how did power vary? What about γ rhythms (see γ burst during working memory by Earl Miller and colleagues)?

We have accordingly conducted additional analyses: the distribution of the vector length as a measure of phase locking was extended and repeated using only cells that were significantly phase locked. For none of the tested conditions, frequency bands 2.5 – 5 Hz, 5 – 12 Hz and 15 – 30 Hz, egocentric or allocentric goal location, was the distribution significantly different from a distribution where the class labels, i.e. the currently remembered goal or the goal in egocentric coordinates were shuffled. We have also extended our decoding analysis using spike counts at specific phases. Here, to increase sensitivity as much as possible, we have included all cells that were significantly phase locked for any of the three goals in allocentric or egocentric coordinates, corresponding to the type of class labels used for decoding. The results were comparable to those without such selection. These new results are presented in the revised manuscript in Figure 5—figure supplement 1.

The frequency bands chosen for analysis were those that exhibited the highest power across the delay period. We should note that there was an offset error in the time windows for the LFP analysis in the original manuscript. This has been corrected, and the updated figures are in the manuscript. The updated spectrograms showed a peak of power in the 2.5-5 Hz frequency band in addition to the 5-12 and 15-30 Hz bands, so we also analyzed those data, the results were comparable to those for the other two bands and we have added the new frequency band to the classification analyses figures and added a note referring to the 2.5 – 5 Hz frequency band in the distribution of vector length analysis. We have now also added a plot of the mean power over the delay period next to the mean spectrograms shown in Figure 5 and Figure 5—figure supplement 1.

Regarding γ bursts, to the best of our knowledge, in the work detecting γ bursts during the delay period, the delay periods generally also showed overall working memory content in the firing rates throughout the delay, which was not observed in our data. Therefore, we think that the potential role of γ bursts in supporting working memory in tasks such as ours is a potentially promising avenue for future investigation. Related to this, we did perform the covariance analysis of Barbosa et al., 2020, which is one of the few studies that looks into how stimulus information could be maintained in the absence of differential firing rates, though in our case goal information was not detected.